

# Interlimb and trunk asymmetry in the frontal plane of table tennis female players

Ziemowit Bańkosz[1], Arletta Hawrylak[2], Małgorzata Kołodziej[1], Lenka Murinova[3] and Katarzyna Barczyk-Pawelec[2]

[1] Department of Biomechanics, Wrocław University of Sport and Health Science, Wrocław, Poland
[2] Department of Kinesiology, Wrocław University of Sport and Health Science, Wrocław, Poland
[3] Department of Natural Sciences in Kinanthropology, Palacký University Olomouc, Olomouc, Czech Republic

Corresponding author
Ziemowit Bańkosz,
ziemowit.bankosz@awf.wroc.pl

## ABSTRACT

**Background**. An interesting and little-reported problem in the literature is the scale of asymmetry in table tennis players, the magnitude of which should perhaps be treated as a risk for injury. Determining the degree of asymmetry in table tennis players can indicate the need to appropriately manage the training process, including compensatory or corrective exercises in the training program, especially since recent studies confirm that training interventions can reduce sporting asymmetries and improve performance. This study aimed to assess the amount of asymmetry in the trunk regarding the frontal plane and the difference between limb circumferences in female table tennis players compared to the control group (non-athletes).

**Methods**. Twenty-two women took part in the study. Ten of them were table tennis professionals with an average training experience of $7 \pm 4.3$ years (the exclusion criterion of the study was a minimum of 3 years of training experience). As a comparison group, the study included 12 female students who did not participate in competitive sports. Body posture was assessed in all subjects using equipment for computer analysis of asymmetry in the torso using the photogrammetric method. Additionally, all the subjects had their upper and lower limb circumferences measured.

**Results**. The results of the conducted research showed asymmetry in the frontal plane in the table tennis player group. As many as six parameters—regarding the pelvic rotation angle, angle of trunk inclination, the height of the angles of the lower shoulder blades and their distance from the spine, as well as the waist triangles, difference in the width and height of the waist triangles and the angle of trunk inclination—indicated asymmetry in this group but significantly differed from the control group ($p \leq 0.05$) only in the first parameter given above. The calculated differences in circumference between the right and left sides in the individual groups were statistically different in several cases ($p \leq 0.05$). This concerned the circumferences of the arms, forearms, elbows, and knees of table tennis players.

**Conclusions**. The research carried out in this study allowed us to determine the occurrence of asymmetry in the frontal plane of the trunk and between the limbs of table tennis players. According to some studies, this may be a risk factor for injury. However, despite the lack of uniform views in the literature on the importance and threats resulting from asymmetries, it appears that, if only for aesthetic reasons, table tennis

would require compensatory or corrective training aimed at developing symmetry of the body structure.

# INTRODUCTION

Asymmetry is defined as a lack of symmetry, understood as the exact similarity in size, shape, and form of two halves of the body that are divisible with respect to a selected axis. Human asymmetry is considered a typical and normal symptom of the human body structure (*Zeyland-Malawka & Prętkiewicz-Abacjew, 2006*; *Afonso et al., 2022*). Morphological asymmetry concerns the differences between the weight of the right and left halves of the body, the length, the circumference, and the position of even-numbered body parts (*Zeyland-Malawka & Prętkiewicz-Abacjew, 2006*). It is often perceived as a disruption in the symmetry of the external shapes of the body on both sides of the median plane (*Koszczyc, 1991*). It typically affects the limbs, but also can affect the trunk and head. Morphological asymmetry is often associated with lateralization and functional dominance of one side of the body (*Kozlenia, Struzik & Domaradzki, 2022*). Many authors state that there is no evidence that asymmetry, taking into account the needs of individual sports disciplines, improves or hinders the achievement of high sports results (*Maloney, 2019*; *Afonso et al., 2022*). However, other studies come to different conclusions. *Fox, Pearson & Hicks (2023)* conducted an extensive meta-analysis, which found a negative impact of asymmetry on changing direction and sprint performance but did not confirm this in the case of the high jump. There are many indicators that most athletes, due to the specificity of a given sports discipline, experience significant asymmetries. Examples include the work of *Maloney (2019)* with participants in hockey, long jump, fencing, soccer, and others. *Turner et al. (2016)*, for example, found asymmetry between the lower limbs in young fencers. The results of *Kobayashi et al. (2010)* indicated that lower limb joint torques may be bilaterally asymmetric in long jumpers. *Rouissi et al. (2016)* observed differences in strength between the lower limbs of soccer players. *Chapelle et al. (2022)* found significant asymmetry exists in the circumference and mass of the upper and lower limbs in tennis players, at the same time pointing out that the measurement of functional asymmetry should be carried out independently of the morphological one. *Hart et al. (2016)* assessed the level of asymmetries in soccer players, concluding that asymmetries were evident in athletes as a product of limb function over time. According to *Afonso et al. (2022)*, bilateral asymmetry is typical for sports in general, presenting and adding to the disciplines mentioned in the above research results in the fields of volleyball, basketball, and football, as well as running, cycling, and others. Nevertheless, some data suggests that asymmetry (morphological, functional) may be associated with the risk of injury, and this problem requires further research (*Afonso et al., 2022*; *Fox, Pearson & Hicks, 2023*; *Guan et al., 2022*). *Butler et al. (2013)* researched a group of American football players, concluding that asymmetry in lower limb balance

may be related to the risk of injury in participants. Strength imbalance was determined as a risk factor for the development of non-contact, acute lower extremity injuries in physical education students (*De Blaiser et al., 2021*) and in shoulders of volleyball players (*Zuzgina & Wdowski, 2019*). *Krzykała et al. (2023)* pointed out that athletes (canoeists) with large morphological asymmetries require greater attention to reduce the risk of overload injuries. However, *Afonso et al. (2022)* suggested that there is no evidence of a relationship between bilateral asymmetry and injury risk in sports. *Helme et al. (2021)* stated that, probably due to the differences in the quality of research and the tools used, the relationship between asymmetry and the risk of injury indicated in the studies is at a medium or low level, and further research requires better, more standardized methods.

Table tennis is a discipline in which players use one hand to play. High-speed movements of the lower body, in which fast but short accelerations are combined with braking actions, coordinated with high-speed execution of different techniques performed by the dominant arm, are some of the skills involved in table tennis (*Padulo et al., 2016*). An athlete generally spends between 4–6 h per day at the table, performing multiple repetitions of various exercises and hitting the ball with a racket using one hand. Success in table tennis depends on the level of many components, such as fitness (speed, endurance, flexibility, and motor coordination), technical, tactical, mental preparation, and many others (*Kondrič, Milić & Furjan-Mandić, 2007*; *Hawrylak, Załuski & Hawrylak, 2021*; *Zagatto, Morel & Gobatto, 2010*). Striking movements are often associated with body rotation to one side, with forehand shots characterized by greater body involvement and greater use in point-winning actions. Players perform up to 1,300 shots during a match, moving up to 2,000 m (*Hudetz, 2005*). These multiple repetitions place a local load on a specific group of muscles and body parts, which are perceived as a risk of injury. Some authors have even reported cases of focal task-specific dystonia in table tennis players (*Le Floch et al., 2010*).

An interesting and little-reported problem in the literature is the scale of asymmetry in table tennis players. The finding of a high asymmetry scale may indicate the need for further research regarding its possible relationship with the occurrence of injuries. A large scale of asymmetry may also indicate the need to conduct special compensatory and corrective exercises, not only for health reasons but also for aesthetic reasons. Previous research indicates that, for example, adopting the ready position is also associated with an increase in asymmetry in the position (rotation) of the pelvis and spinous processes (the sagittal plane, *Bańkosz & Barczyk-Pawelec, 2020*). This study demonstrated also the dominance of kyphotic body posture in table tennis players, which can be caused by many hours of using the ready position during playing. The above asymmetries and body postures may cause overload in the spine, which in turn may cause pain. However, there are no research results assessing asymmetry of the trunk in the frontal plane or any data on the occurrence of differences in the structure of limbs in athletes practicing in table tennis, a discipline in which most of the athlete's activities are asymmetric. It can be hypothesized that table tennis players have a higher level of interlimb and trunk asymmetry than people who do not practice sports. The finding of asymmetry of the trunk or interlimb areas may indicate the presence of threats related to the risk of overload injuries, as reported in the literature in relation to other sports disciplines. Determining the magnitude of asymmetry in table

tennis players will also indicate the need to appropriately manage the training process, including compensatory or corrective exercises in the training program, especially since recent studies confirm that training interventions can reduce sporting asymmetries and improve performance (*Maloney, 2019*). Therefore, this study aimed to assess the amount of trunk asymmetry in the frontal plane and the difference between limb circumferences in female table tennis players compared to a control group (non-athletes), as well as to compare the differences between left and right limbs in both groups. Taking into account literature reports and the specificity of table tennis, we hypothesize that female table tennis players show greater asymmetry than the comparison group.

## MATERIALS & METHODS

### Participants

Twenty-two young women took part in the study. Ten of them practiced table tennis professionally in local, provincial sports centers (six times a week, and some of them more often—twice a day) with an average training experience of $9.3 \pm 3.3$ years. Five of them represented their clubs playing in the highest division in women's team competition. Two of these were Chinese players, seven others participants were members of national seniors or juniors women's team. As a comparison group, the study included 12 female students from the local high school who stated they do not practice competitive sports. The exclusion criteria from the study were a minimum of 3 years of training experience (only for the training group), current injuries, and chronic diseases (for both groups). The characteristics of the study participants are presented in Table 1. All tests were performed in the laboratory in the morning and were carried out between 2021–2022. All the subjects were informed about the purpose and course of the research, and signed informed consent to participate in the study was obtained. The Senate's Research Bioethics Commission approved this research study at the Wrocław University of Health and Sports Sciences.

### Research method

Body posture was assessed in all subjects using equipment for computer analysis of asymmetry of the torso using the photogrammetric method based on the Moiré phenomenon (*Barczyk-Pawelec, Bańkosz & Derlich, 2012*; *Barczyk-Pawelec & Sipko, 2017*; *Drzał-Grabiec et al., 2013*, http://cq.com.pl/e_pl_podstawy.html). In medicine, the Moiré method is a process of 3-dimensional morphometry, in which contour maps (photograms) are produced on the body surface from overlapping interference fringes created when a human body is illuminated by beams of coherent light coming from two different point sources (*Domagalska, Szopa & Lembert, 2011*). The measurement station consists of a computer with a card, program, monitor, printer, projection, and a reception device with a camera for measuring the back and feet (Fig. 1). Obtaining a three-dimensional image is possible by displaying lines with precisely defined parameters on the patient's back. Lines falling on the back are distorted depending on the surface configuration. Thanks to the use of a lens, the subject's image can be received by a special optical system using a camera and then transmitted to a computer monitor. The distortions of the line image recorded in the computer's memory are converted by a numerical algorithm into

**Table 1  Characteristics of groups studied.**

|  | Table tennis players $N = 10$ | Controll group $N = 12$ | $p$ | Cohen's d |
|---|---|---|---|---|
| Age (years) | $17.2 \pm 2.7$ | $17.5 \pm 1.24$ | 0.895 | 0.15 |
| Body mass (kg) | $55 \pm 10.48$ | $56.67 \pm 11.66$ | 0.843 | 0.15 |
| Body height (cm) | $164.3 \pm 6.53$ | $165.75 \pm 7.35$ | 0.921 | 0.21 |

**Notes.**
The differences are significant with $p < 0.05$ and effect size Cohens $d \geq 0.8$.

a contour map of the examined surface. The obtained image of the back surface allows for a multi-aspect interpretation of body posture. In addition to assessing the symmetry of the torso, it is possible to determine the angle of rotation of the vertebrae, the size of the rib hump, and the measurement of spinal curvature, *i.e.,* the distance of the top vertebra from the C7–S1 line (*Mrozkowiak, 2007*). Based on the stored image of the torso in the entered data of the examined persons, three-dimensional coordinates of the body surface were obtained, and parameters determining the amount of asymmetry in the shoulder and hip girdle and the inclination of the torso in the frontal plane were calculated (Fig. 2). Analysis was performed using computer software (CQ Electronic System, Poland). A detailed description of the calculation procedure can be found in the manufacturer's information, http://cq.com.pl/mora4g.html). The reliability, validity, advantages, and limitations of the photogrammetric method have been discussed in the literature (*Mrozkowiak & Strzecha, 2012*; *Drzał-Grabiec et al., 2013*; *Labecka & Plandowska, 2021*; *Carneiro et al., 2014*). Many studies found this method to have high repeatability and high intraobserver and interobserver correlation. Correlation between photogrammetric parameters and radiographic Cobb angles ranged from moderate to high (*Labecka & Plandowska, 2021*; *Saad et al., 2012*). Photogrammetric accuracy is estimated to be 94% (*Drzał-Grabiec et al., 2013*). Before starting the examination, the following points were marked on the subject's body with a black, washable marker: the spinous processes of the spinal vertebrae from C7 to S1 and the thoracolumbar transition, the acromion processes and the lower angles of the scapulae, and the posterior superior iliac spine. Body posture was assessed according to methods previously described in the literature (*Bańkosz & Barczyk-Pawelec, 2020*; *Barczyk-Pawelec et al., 2022*). The subject stood in a free-standing, relaxed position, without shoes in the camera's field of view at a distance of 2.6 m (Fig. 1). The subject's feet were placed on a line parallel to the measurement station, hip-width apart. The knee joints were straight, and the body weight was evenly distributed on both lower limbs. The upper limbs were positioned freely along the body, the head was positioned freely, and the gaze was directed straight ahead. After adopting such a free, habitual posture, the image of the back was registered.

All the measurements were taken by the same investigator at the same time of day (early afternoon hours) and in similar conditions (a dark room with controlled ambient temperature). The investigator was a very experienced physiotherapist who deals with photogrammetric tests on a daily basis. Spinal posture was recorded continuously for 3 s at 4 Hz to capture twelve images, from which the sixth image was extracted for subsequent

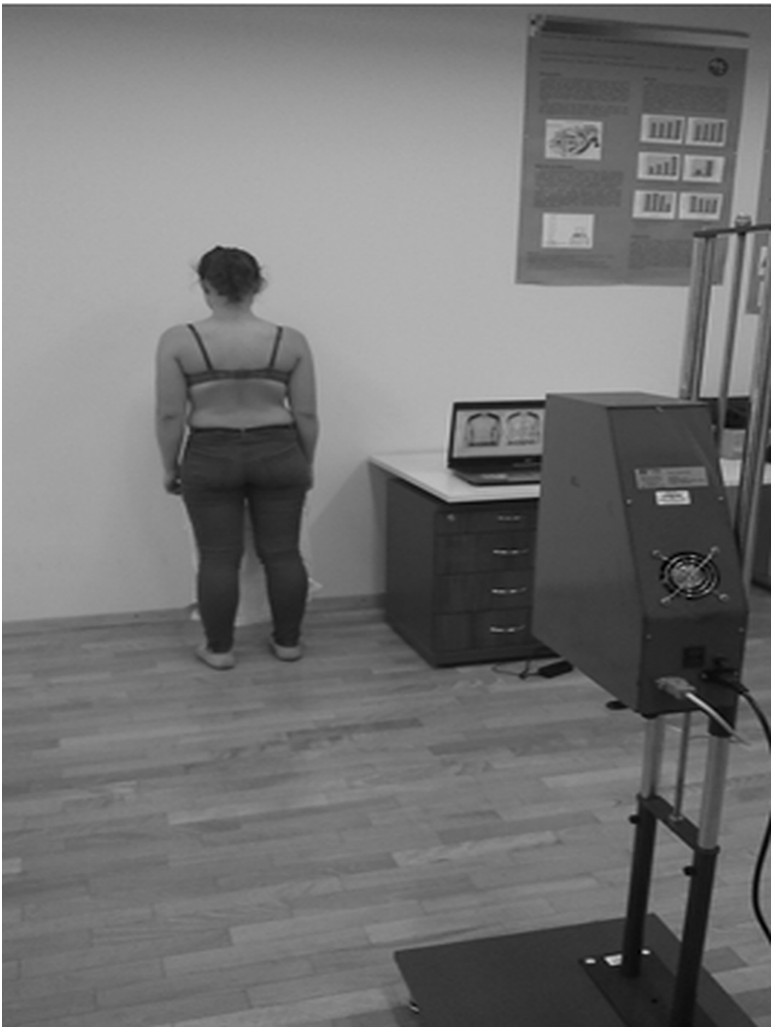

**Figure 1  The research station.**

analysis. The participants were not aware of when the recording was performed (*Bańkosz & Barczyk-Pawelec, 2020*; *Barczyk-Pawelec et al., 2022*). Further analysis took place without the presence of the examined person. Using computer software, a report was obtained with parameters that were subject to further analysis.

In the frontal plane, the following torso asymmetries were assessed and analyzed:

(a) Angular parameters (expressed in degrees):

- KNT - angle of trunk inclination,
- KLB - angle of shoulder line inclination,
- KNM - pelvic inclination angle, and
- KSM - pelvic rotation angle.

(b) Length and depth parameters (expressed in mm):

- UL - difference in the positions of the inferior angles of the scapula,
- OL - difference in the distance of inferior angles of the scapula from the spine,

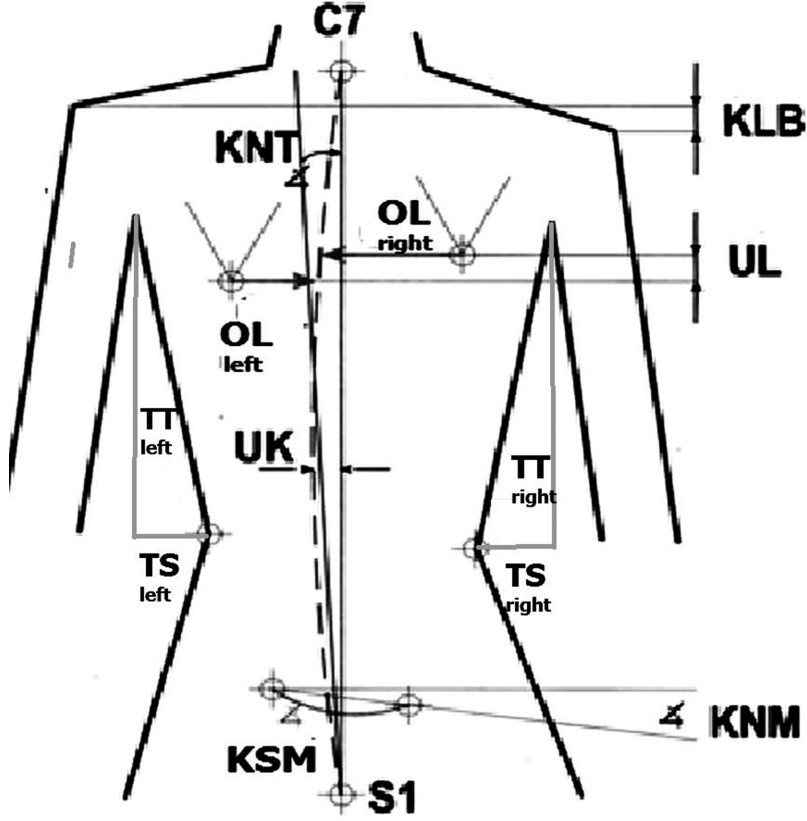

**Figure 2** The examined parameters—photogrametric method.

- TT - difference in the height of the waist triangles,
- TS - difference in the width of the waist triangles, and
- UK - deviation of spinous processes from the line of the spine.

The magnitude of asymmetry was established based on differences in the placement of osteal points within the trunk. Intervals of these differences were determined for angular and length parameters, distinguishing three levels of asymmetry according to *Bibrowicz (1995)*.

For the angle indices (KNT, KLB, KNM, KSM), it was assumed that:
- the difference of $0° < x \leq 1.5°$ means no asymmetry,
- the difference of $1.5° < x < 3°$ means moderate asymmetry, and
- the difference of $x \geq 3°$ indicates severe asymmetry.

For linear asymmetry indices (UL, OL, TT, TS, UK), it has been assumed that:
- the difference of $0 < x \leq$ five mm means no asymmetry,
- the difference of $5 < x < 10$ mm means moderate asymmetry, and
- the difference of $x \geq 10$ mm means severe asymmetry.

Additionally, all the subjects had their upper and lower limb circumferences measured. The measurements of limb circumferences were made with a centimeter tape with an accuracy of 0.5 cm. The centimeter tape was applied perpendicular to the long axis of the

limb at the following levels: the thickest place in the arm, the elbow joint, the thickest place in the forearm, the wrist joint, the thickest place in the thigh, the knee joint, the thickest place in the lower leg, and at the ankle joint. The measurement position was a standing, free position, with upper limbs placed along the body, and the load on both legs being equal. An experienced physiotherapist performed the examination. Every measurement of circumference was repeated twice to make sure it was correct. After taking the measurements, the limb circumferences of the training and non-training groups were analyzed. Differences between the circumferences of the individual's right and left limbs were calculated as indicators of interlimb asymmetry. Using this indicator, both groups were compared.

**Statistical analysis**
The means and standard deviations were calculated for all variables. The normality of distribution was verified with the Shapiro–Wilk test and the homogeneity of variances with Levene's test. Differences between the training and non-training groups were assessed with the Student's $t$-test, with independent variance estimation if the condition of homogeneity was not met, and in the absence of normal distributions with the Mann–Whitney U test. Cohen's d for the difference in means in independent samples was used to assess the effect size of differences. At $0.2 \leq d < 0.5$, the effect size was assessed as small, at $0.5 \leq d < 0.8$ as medium, and $d \geq 0.8$ as large (*Cohen, 1988*). For Cohen's $d = 1.0$ and sample size $n = 22$, the estimated power of the tests was 0.90. Comparisons between the right and left sides of measurements, taking into account groups, were performed using two-way analysis of variance and Tukey's *post-hoc* test (HSD), and if necessary, the non-parametric Kruskal-Wallis test was used. In the multiple analysis, the type of the studied group (players –control) and the side of the limb (right-left limb) were taken into account. Statistical significance of the results was assumed at $p < 0.05$. Statistical analyses were performed using TIBCO Statistica® 13.3 (StatSoft Polska).

# RESULTS

Table 2 presents the results characterizing both groups in relation to the size of asymmetry parameters in the trunk area and the differences between the group of tennis players and the non-training group. When assessing the amount of asymmetry in the torso area, it should be concluded that the group of table tennis players was characterized by high asymmetry in the following parameters: KSM, UL, OL, TT, and TS, moderate asymmetry in the KNT parameter, and no asymmetry in KLB, and UK. The control group was characterized by no asymmetry in the KNT, OL, TS and UK parameters. A high asymmetry in control group was found only with regard to the TT parameter, and medium (moderate) asymmetry in every other parameter (KLB, KNM, KSM and UL). Significant differences between the groups were found in relation to the KSM parameter—this parameter was larger in the table tennis group, as well as KNM and UK (smaller parameters in the table tennis group—Table 2).

Concerning the size of the measured circumferences (circ.—Table 3), it was found that both groups had similar sizes compared to the same sides of the body, except the right

**Table 2 The values of spine examination parameters.**

|  | Table tennis players $N = 10$ | Controll group $N = 12$ | $p$ | Cohen's d |
|---|---|---|---|---|
| KNT (deg) | 1.58 ± 0.64[MA] | 1.00 ± 0.64 | 0.056 | 0.91 |
| KLB (deg) | 1.22 ± 0.84 | 1.53 ± 0.96[MA] | 0.468 | 0.34 |
| KNM (deg) | **1.03 ± 1.06** | **2.79 ± 1.51[MA]** | **0.010** | **1.33** |
| KSM (deg) | **3.71 ± 2.34[HA]** | **1.67 ± 1.71[MA]** | **0.023** | **1.01** |
| UL (deg) | 3.02 ± 1.63[HA] | 2.73 ± 2.39[MA] | 0.468 | 0.16 |
| OL (mm) | 13.18 ± 11.64[HA] | 9.89 ± 7.17 | 0.598 | 0.35 |
| TT (mm) | 10.95 ± 7.56[HA] | 13.02 ± 16.43[HA] | 0.742 | 0.16 |
| TS (mm) | 14.99 ± 9.51[HA] | 8.99 ± 5.16 | 0.138 | 0.81 |
| UK (mm) | **3.27 ± 1.83** | **5.37 ± 2.47** | **0.041** | **0.95** |

Notes.
The differences are significant with $p < 0.05$ and large effect size with Cohens $d \geq 0.8$ (bold).
KNT, angle of trunk inclination; KLB, - angle of shoulder line inclination; KNM, pelvic inclination angle; KSM, pelvic rotation angle; UL, difference in the positions of the inferior angles of scapula; OL, difference in the distance of inferior angles of scapula from the spine; TT, difference in the height of the waist triangles; TS, difference in the width of the waist triangles; UK, deviation of spinous processes from the line of the spine; MA, moderate asymmetry; HA, high asymmetry.

wrist, the circumference of which was significantly larger in table tennis players. However, the calculated differences in circumference between the right and left sides (d_circ) in the individual groups were statistically different in several cases. This concerned the differences in circumferences of the arm, forearm, elbow, and knee. A large effect size ($d \geq 0.8$) was obtained for all confirmed differences.

In the multiple analysis, taking into account the type of the studied group (players–control) and the side of the limb (right-left limb), no main effect of the group was found to differentiate limb circumferences. It was found that the side of measurement and its interaction with the type of group were significant effects differentiating: arm circumference ($F = 10.928$, $p = 0.004$; $F = 6.126$, $p = 0.022$, respectively for the effect of side of measurement and the R-L *vs.* group interaction), forearm ($F = 17.323$, $p < 0.001$; $F = 7.015$, $p = 0.015$, for the side effect and interaction, respectively) and elbow ($F = 11.853$, $p = 0.003$; $F = 5.088$, $p = 0.035$, for the side effect and interaction, respectively). There were significant differences between the right and left circumferences indicated above only in the training group (*post-hoc* HSD test: $p = 0.004$, $p = 0.001$, $p = 0.005$ for arm, forearm, and elbow circumferences, respectively). No differentiating effects were confirmed for the other circumferences.

## DISCUSSION

The aim of the research carried out in this study was to assess the amount of asymmetry of the trunk in the frontal plane and the difference between limb circumferences in table tennis players compared to a control group (non-athletes).

### Trunk asymmetries in the frontal plane
The results of the conducted research showed asymmetry of the trunk in the frontal plane in the table tennis players group. As many as five parameters—regarding the pelvic rotation

**Table 3  Values of limb circumferences and differences between left and right limbs.**

|  | Table tennis players<br>N = 10 | Controll group<br>N = 12 | p | Cohen's d |
|---|---|---|---|---|
| circ. Arm R | 25.35 ± 1.68 | 25 ± 2.66 | 0.553 | 0.15 |
| circ. Arm L | 23.9 ± 1.96 | 24.79 ± 3.09 | 0.468 | 0.34 |
| circ. Forearm R | 22.95 ± 1.46 | 22.17 ± 1.71 | 0.199 | 0.49 |
| circ. Forearm L | 21.45 ± 1.54 | 21.83 ± 1.79 | 0.468 | 0.23 |
| circ. Elbow R | 23.6 ± 1.82 | 22.79 ± 1.47 | 0.277 | 0.49 |
| circ. Elbow L | 22.4 ± 1.31 | 22.54 ± 1.27 | 0.792 | 0.11 |
| circ. Wrist R | **15.75 ± 0.82** | **14.88 ± 0.86** | **0.030** | **1.03** |
| circ. Wrist L | 15.3 ± 1.01 | 14.71 ± 0.84 | 0.199 | 0.64 |
| circ. Thigh R | 48.4 ± 3.71 | 50.63 ± 6.13 | 0.553 | 0.43 |
| circ. Thigh L | 47.55 ± 4.04 | 50.33 ± 5.66 | 0.323 | 0.56 |
| circ. Shank R | 34.5 ± 2.22 | 36.17 ± 4.21 | 0.277 | 0.48 |
| circ. Shank L | 34.2 ± 2.5 | 36.17 ± 3.93 | 0.210 | 0.59 |
| circ. Knee R | 34.85 ± 3.03 | 34.71 ± 2.81 | 0.692 | 0.05 |
| circ. Knee L | 34.55 ± 2.45 | 34.46 ± 2.97 | 0.843 | 0.03 |
| circ Ankle R | 22.25 ± 1.75 | 22.21 ± 1.47 | 0.598 | 0.02 |
| circ. Ankle L | 22.1 ± 1.52 | 21.96 ± 1.64 | 0.999 | 0.09 |
| d_circ. Arm. | **1.75 ± 0.79** | **0.63 ± 0.86** | **0.005** | **1.35** |
| d_circ. Forearm | **1.8 ± 0.63** | **0.75 ± 0.45** | **0.001** | **1.95** |
| d_circ. Elbow | **1.6 ± 0.81** | **0.25 ± 0.45** | **0.001** | **2.12** |
| d_circ. Wrist | 0.65 ± 0.53 | 0.25 ± 0.4 | 0.093 | 0.86 |
| d_circ. Thigh | 0.95 ± 0.8 | 0.79 ± 0.72 | 0.692 | 0.21 |
| d_circ. Shank | 0.4 ± 0.39 | 0.75 ± 0.72 | 0.307 | 0.59 |
| d_circ. Knee | **1 ± 0.78** | **0.33 ± 0.39** | **0.016** | **1.12** |
| d_circ Ankle | 0.45 ± 0.37 | 0.33 ± 0.33 | 0.510 | 0.34 |

**Notes.**
The differences are significant with $p < 0.05$ and large effect size with Cohens $d \geq 0.8$ (bold).
circ, circumference; d_circ, difference in circumference.

angle (KSM), the height of the angles of the lower shoulder blades and their distance from the spine (UL and OL), as well as the waist triangles (TT and TS)—indicated high values of asymmetry in this group, and one (KNT)—moderate. Compared to non-training people, who were also characterized by asymmetry in most measurements but at a "moderate" level, it can be seen that the group of competitive women is characterized by a bigger number of parameters that indicated the occurrence of high asymmetry of the trunk in the frontal plane, although a statistical difference was found only in the KSM parameter. Therefore, it should be stated that the studied group of female table tennis players shows only a slightly higher degree of asymmetry than the control group. However, in both groups, asymmetry in the trunk is visible. The huge angle of pelvic rotation (KSM) in table tennis players may result from a permanent, long-lasting playing position, which is often taken with one leg extended forward. The vast number of rotational movements of the torso occurring during punches must also be significant (*McAfee, 2009*). As mentioned, there are no uniform views in the literature on the occurrence of asymmetry in athletes and

their impacts on the sport's results. Many believe that increasing morphological asymmetry is typical of sports and is not related to the possibility of limiting the achievement of a high sports level (*Afonso et al., 2022*). However, it should be remembered that there are diverse results in the literature, and other conclusions can also be found (*Fox, Pearson & Hicks, 2023*). It should be noted that asymmetries in structure may, according to many, lead to injuries and pain (*Lijewski et al., 2021*; *Bańkosz & Barczyk-Pawelec, 2020*). *Grabara (2014)*; *Grabara (2015)* stated that asymmetrical spinal overload, which often occurs in sports training, should be monitored by meticulous assessment of body posture in young athletes. It should be also added that the UK and KNM results showed significantly higher values in the control group than in the table tennis players. However, in both groups the value of the UK parameter indicated the lack of asymmetry. But the value of the KNM parameter indicated moderate asymmetry in control group and no asymmetry in athletes. It should be added that the UK and KNM results showed significantly higher values in the control group than in the table tennis players. However, in both groups the value of the UK parameter indicated the lack of asymmetry. But the value of the KNM parameter indicated moderate asymmetry in control group and no asymmetry in athletes. Similar differences or their lack between sportsmen and no-sportsmen in the values of this parameter can be found in the literature on female track and field and volleyball players (*Grabara & Hadzik, 2009*; *Grabara, 2015*). It is possible that the quite large number of asymmetries also in the control group reflects the occurrence of asymmetries in the so-called normal population. The high asymmetries (KNM, KNT, OL, TT, TS), and in one case even larger than in non-training group (KSM), in table tennis players shown in this study indicate that the scope of measures used in training programs should be expanded to include compensatory or corrective exercises and the size of the asymmetry should be monitored. The described asymmetries also, in light of the previously cited literature, could pose a risk for injury to the spine or trunk. Previous research (*Bańkosz & Barczyk-Pawelec, 2020*) showed, for example, that spine pain reported by table tennis players was related to excessive load on the spine and the kyphotic posture often occurring in the subjects. *Maloney (2019)* claims that recent investigations have demonstrated that training interventions can reduce sporting asymmetries and improve performance. These interventions could be: bilateral training, targeted unilateral training, balance exercises, resistant training or individualized rehabilitation (*Maloney, 2019*). Therefore, such effects can also be expected in table tennis. The results obtained in this study draw attention to the substantial asymmetry in the OL parameter among table tennis players. Similar asymmetries of shoulder blade positioning were found in tennis players by *Oyama et al. (2008)*. These authors stated that asymmetry should be considered a normal phenomenon resulting from using one limb in a specific way. Research results indicate that asymmetries within the trunk and asymmetries between limbs may have different effects and impacts on the health and sports level of an athlete, so they probably need to be considered and monitored separately.

## Differences in the circumferences of limbs

The differences in the circumferences of the left and right limbs assessed in this study (adopted in this paper as a measure of interlimb asymmetry, d_circ) were also larger in

the examined women who practice table tennis than in those who do not. It is associated with larger circumferences on the playing side than on the non-playing side, particularly in the arm, forearm, and at the elbow and knee levels. This undoubtedly results from using a given limb to play, *i.e.,* repeated actions performed with it. Similar observations can be found in the literature in relation to other sports, such as football, rhythmic gymnastics, and field hockey (*Hart et al., 2016*; *Frutuoso et al., 2016*; *Krzykała & Leszczyński, 2015*). *Afonso et al. (2022)* state that bilateral asymmetries are prevalent in sports, do not seem to impair performance, and no evidence suggests that they increase injury risk. *Fox, Pearson & Hicks (2023)*, reviewing works on the differences between limbs in athletes and the importance of these differences in achieving sports results, found that in terms of some elements of sports mastery, these asymmetries and differences may also have a negative impact. However, as previously written, there are many reports indicating the risk of injury in sports caused by asymmetry (functional but also morphological). This is indicated in the research by *Krzykała et al. (2023)*, *Kozlenia, Struzik & Domaradzki (2022)*, *De Blaiser et al. (2021)*, and others *Zuzgina & Wdowski (2019)*. Also, *Steidl-Müller et al. (2018)* stated that limb differences in unilateral leg extension strength represent a significant injury risk factor in youth ski racers.

## The risk of injury and asymmetries in table tennis

The literature states that table tennis as a sport is characterized by a low rate of sport injuries compering with other sports (*Biz et al., 2022*; *Ko et al., 2023*). However, lumbago and lower back pain are very common among them (*Biz et al., 2022*). It is possible that the frequent cases of back pain in table tennis indicated also in the previous work (*Bańkosz & Barczyk-Pawelec, 2020*) are also related to asymmetry in the spine. In light of these statements, it can be concluded that the asymmetries in table tennis players in the trunk area may also indicate a risk for injury in this sports discipline. Although the findings presented in the literature may be interpreted as not fully convincing (*i.e.,* asymmetries and risk of injury), some of the asymmetries demonstrated in the present research may constitute the basis for further research in this area. *Correa-Mesa & Correa-Morales (2020)* found that the shoulder, knee, back and elbow were parts of the body the most affected by injuries (*Biz et al., 2022*). The same authors stated that the most prevalent type of injury was tendinopathy, benign muscle injuries and sprains. Perhaps the disproportions between limb circumferences found in this study also could pose a risk of injuries and overloads in the above-mentioned body areas, like shoulder, elbow or knee. Also, *Folorunso, Mutiu & Ademola (2010)* in their research highlighted the danger of the occurrence of 'the rotator cuff impingement syndrome' in table tennis players due to muscle hypertrophy on the playing side. This does not change the fact that it can also be assumed that in table tennis, the phenomenon of increasing the circumference of the upper limb on the playing side is a typical expression of adaptation to the activities performed, which is also noted in the literature on table tennis (*Munivrana, Paušić & Kondrič, 2011*). However, since, as previously mentioned, the views of the importance of these asymmetries for achieving sports mastery and the risk of injury vary, this issue should be approached carefully and

undoubtedly requires further research. Nevertheless, it seems that for aesthetic reasons, table tennis would require compensatory or corrective training aimed at body symmetry.

## Limitations of the study

Undoubtedly, the research undertaken in our work has its limitations. First of all, we only studied groups of women. To generalize the obtained results and conclusions, the research should be supplemented with groups of men. It would undoubtedly also be necessary to expand the scope of research to include a larger number of highly qualified players. Perhaps the assessment and comparison of the research results on the occurrence of asymmetry should be extended to athletes of different sports advancement. Our research concerned only female athletes with a high level of sport. It must also be admitted that our assessment of interlimb asymmetry concerned absolute differences between limb circumferences, but we were unable to assess its size (large, small, *etc.*) due to the lack of such data in the literature. A certain limitation of our work is the low resolution of the tape measurement, which was 0.5 cm.

# CONCLUSIONS

The research carried out in this study allowed us to determine the occurrence of asymmetry of the trunk in the frontal plane and between limbs in table tennis players. In several cases asymmetries were high, in one examined parameter the table tennis group differed significantly from the control group. Most likely, these asymmetries result from the specificity of the discipline, in which asymmetric positioning in relation to the table and one-sided work seem to be the most critical factors. These asymmetries may result, according to the findings of other studies available in the literature, from the processes of adjustment and adaptation to specific training, and may also pose a risk of injury. Despite the lack of uniform views in the literature on the importance and threats resulting from these asymmetries, it appears that, if only for aesthetic reasons, table tennis players may require compensatory or corrective training aimed at the symmetry of the body structure.

## Funding

The authors received no funding for this work.

## Competing Interests

The authors declare there are no competing interests.

## Author Contributions

- Ziemowit Bańkosz conceived and designed the experiments, performed the experiments, analyzed the data, prepared figures and/or tables, authored or reviewed drafts of the article, and approved the final draft.
- Arletta Hawrylak conceived and designed the experiments, performed the experiments, prepared figures and/or tables, authored or reviewed drafts of the article, and approved the final draft.

- Małgorzata Kołodziej conceived and designed the experiments, performed the experiments, analyzed the data, prepared figures and/or tables, authored or reviewed drafts of the article, and approved the final draft.
- Lenka Murinova conceived and designed the experiments, performed the experiments, authored or reviewed drafts of the article, and approved the final draft.
- Katarzyna Barczyk-Pawelec conceived and designed the experiments, performed the experiments, analyzed the data, prepared figures and/or tables, authored or reviewed drafts of the article, and approved the final draft.

## Human Ethics

The following information was supplied relating to ethical approvals (i.e., approving body and any reference numbers):

The Senate's Research Bioethics Commissionof Wrocław University of Sport and Health Science approved the research.

## Data Availability

The raw measurements are available in the Supplementary File.

## Supplemental Information

Supplemental information for this article can be found online at http://dx.doi.org/10.7717/peerj.17526#supplemental-information.

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
