# Peer review of "Interlimb and trunk asymmetry in the frontal plane of table tennis female players"

_PeerJ, doi:10.7717/peerj.17526_

## Round 0.1 · original submission · Major Revisions

Thank you for considering PeerJ to submit your manuscript. Please attend to the reviewer´s comments.

Regards

Dr. Manuel Jiménez

Reviewer 1 ·

Basic reporting

The article has a correct structure, containing 2 figures and 3 tables. Table 1 could be omitted as it merely provides supplementary information about the age, height, and body mass of the study group.
If the study was conducted solely on women, this should be indicated in the abstract – use “women” instead of “people”. Please specify whether the individuals in the control group were also women. If so, I suggest adding “female students”. The authors only mention “students” without specifying gender.
On lines 311-313, I suggest specifying which particular training methods can reduce asymmetries and improve performance.

Experimental design

In my opinion, the manuscript aligns with the aims and scope of the journal. The work addresses the issue of asymmetry in sports. The objective is well-formulated, however, a research hypothesis is missing. The research methods are well-described.
The authors provided a good characterization of the athletes’ group, but little is known about the women (if they were women) in the control group. Were there any exclusion criteria for women in the control group?
How was the effect size calculated if non-parametric tests were used? Did all variables have a normal distribution?

Validity of the findings

The authors included 22 women in their study, with 10 female athletes in the training group. This is not a large sample size, especially considering their varying training experience. Please provide justification for the sample size.

Annotated reviews are not available for download in order to protect the identity of reviewers who chose to remain anonymous.

Reviewer 2 ·

Basic reporting

Line 97-99: The authors states "The aerobic system seems to be the predominant energetic
pathway for activities during the match, but the anaerobic system is very important during effort
periods (Zagatto et al., 2010)" The phrase seems out of context, since it does not contribute to the development of the introduction. I would recommend removing it

Experimental design

Line 187 or 223: please specify if it was a blind evaluator. In case it wasn´t, do not make any changes.

Validity of the findings

The authors point out in the discussion (line 286) a series of asymmetries in the frontal plane of the trunk (KSM, UL, OL, TT and TS) in the group of female athletes. They later point out that asymmetries can lead to pain and injuries, but this is not explored further. I would recommend considering the following aspects: what are the most common injuries in table tennis? what are the posible causes/consequences (muscle shortening, greater compression or joint stressof the asymmetries observed in this study )? Could be these causes related to the most common injuries?.

The authors point out, in both the introduction and the discussion, that the literature is diverse regarding whether asymmetries are adaptations or represent a risk of injury; It would be interesting if they enriched the discussion by relating their results to either adaptations or risk, and pointing to the need for prospective studies that would allow them to reinforce their results.

Line 337- 341: The authors states "In light of these statements, it can be concluded that the asymmetries in table tennis players indicated in our study in the trunk area may also indicate a risk for injury in this sports discipline....". However, the authors title the section as "Differences in the circumferences of limbs". Please change title or reorganize information.

Line 367 - 369: Thue authors states "In light of some findings presented in the literature, it
can be concluded that the asymmetries in table tennis players indicated in our study may also
indicate a risk for injury in this sports discipline." I think the article does not allow us to conclude that. In fact, the discussion is presented in an ambiguous way. I believe that the recommendations made previously would help the authors reinforce this conclusion to be included in the article.

Additional comments

Thank the authors for their work. This is methodologically well structured and easy to understand.

---

## Round 0.2 · Minor Revisions

Dear Author:

Thank you for your patience and for thinking of having PeerJ publish your paper. It is nicely written. I'm concerned about some parts of your manuscript. One reviewer considered rejecting it, and two of them had minor reviews. In my opinion, the sample is short, and you must introduce a power effect to reduce bias, especially on type II errors. Some comments from rejecting reviewers are important to consider in your manuscript to improve the final draft. So, please try to answer these recommendations.

Sincerely,
Dr. Manuel Jiménez

Reviewer 3 ·

Basic reporting

no comment

Experimental design

no comment

Validity of the findings

no commnet

Additional comments

Dear Authors,
the Authors did a good job in writing a readable manuscript. This paper could be considered qualified to be published on PeerJ if the authors apply the modifications requested, "Minor issues" are recommended. It is the reason why "Minor revision" is my personal choice.

The main aim of the submitted manuscript was to assess the amount of asymmetry in the trunk regarding the frontal plane and the difference between limb circumferences in female table tennis players compared to the control group (non-athletes).

This paper could be considered qualified to be published if the authors apply the modifications requested.
While it is a very interesting topic, some suggestions for improving the paper are provided below:

The introduction is very well written including the most important literature about asymmetry in sport. The asymmetries may indicate a risk of injury in many sports disciplines, and it is correctly reported in the manuscript by Authors. Moreover, the relationship between asymmetries and risk of injuries has been treated with caution by all the cited Authors. Indeed, the following words were reported by literature to better describe this relationship: “should perhaps be treated as a risk for injury”, “may indicate a risk of Injury”, “may be related to the risk of injury”, “there is no evidence of a relationship between bilateral asymmetry and injury risk in sports”, “which are perceived as a risk of injury”, “no evidence suggests that they increase injury risk”, etc.
Therefore, I would suggest rewriting some parts of the present manuscript (abstract, Introduction Line 106-107, paragraph Line 341, etc.) in order to be more cautious in connecting “interlimb and trunk asymmetries in the frontal plane of table tennis players” and “risk of injuries”. In other words, this paper is limited to a laboratory measurement to describe the participants' interlimb and trunk asymmetries.

Finally, this manuscript could be considered qualified to be published if the authors apply the previous modifications, therefore “Minor revision” is recommended.

·

Basic reporting

The article has a well-structured format, and the figures and tables are properly formatted (well labelled & described).

Also, it may be worth considering changing the title to "Female Table Tennis Players," as there are significant differences in many aspects between male and female table tennis players in general.

On lines 95-97, there is a sentence: “An athlete training professionally devotes approximately six hours a day to performing multiple repetitions of various exercises and hitting the ball with a racket using one hand”.
Is there a reference for this data, or perhaps it would be more flexible to write that player generally spend between 4-6 hours per day at the table? It is not specified that all players, including professional ones, spend 6 hours on the table.

In line 98, I believe that the term "fitness" or "conditioning preparation" would be more appropriate than "motor preparation."

Experimental design

The manuscript aligns well with the aims and scope of the journal as it addresses the issue of asymmetry in sports. The objective is also well-formulated.

The research methods are well described, and the applied statistical analyses are adequate. I suggest replacing the subtitle "Statistical calculations" with "Statistical analyses."

Validity of the findings

My opinion is that the authors should provide a more precise assessment of the performance levels of female table tennis players, because the data: “Ten of them practiced table tennis professionally in local, provincial sports centers (6 times a week, and some of them more often - twice a day) with an average training experience of 7 ± 4.3 years”, does not indicate their level of success at the national or international level, which ultimately affects the final conclusions.

In line 366-367 authors make a statement: “Our research concerned only female athletes with a high level of sport”. Success in high-level sports is not necessarily determined by the number of hours you practice each day, but rather by your performance in competitions.

Reviewer 5 ·

Basic reporting

In a technical sense the manuscript is professionally written, meaning it is well-structured, both in total and single chapters, clear and easy to follow with an adequate level of language skills used throughout the paper.

The authors provided sufficient context for the topic introduction but the article would benefit if they have included some existing, more sport-specific references of the topic.. e.g.

Folorunso, O., Mutiu, A., & Ademola, O. (2010). The playing posture, activities and health of the table tennis player. International Journal of Table Tennis Sciences, 6, 99–104.
Munivrana, G., Paušić, J., & Kondrič, M. (2011). The incidence of improper postural alignment due to the influence of long-term table tennis training. Kinesiologia Slovenica, 17(2)
Iordan, D., Mereuță, C. and Mocanu, M. (2020) “Aspects of the postural alignment and plantar structure in junior female table tennis players”, Annals of “Dunarea de Jos” University of Galati. Fascicle XV, Physical Education and Sport Management, 2, pp. 2-11.
Biz C, Puce L, Slimani M, Salamh P, Dhahbi W, Bragazzi NL, Ruggieri P. Epidemiology and Risk Factors of Table-Tennis-Related Injuries: Findings from a Scoping Review of the Literature. Medicina. 2022; 58(5):572. https://doi.org/10.3390/medicina58050572

... especially to put findings of their research in the context of previously conducted researches within the specific field.

Provided Figures and Tables are relevant for the content of the article, appropriately described and labelled, but resolution of the Figure 2. (Measured parameters) should be better.

Experimental design

The topic of the paper could be considered as one to fall within the scope of the journal.
However, although the research question of the study is well-defined and comprehensible, its relevance is not very high.
There are no any concrete evidence in the existing scientific literature that a slight asymmetry, which is normal occurrence even in normal population, not to mention unilateral sports, improves or hinders sport achievements or even significantly increase a risk for injuries. So, it is not completely clear what is the ultimate scientific rationale for conducting this research, which knowledge gap the authors have aimed to fill, and which kind of practical implications the research has been expected to provide.
Of course, if a high level of asymmetry in the trunk region exists, an improper postural alignment in a worst case scenario can potentially point to some spinal disorders as scoliosis. One can agree that table tennis, due to the sport's structural characteristics, which are observable in its highly asymmetrical nature (the repetitive performance of strong, one-arm strokes);strong rotational forces that occur in the hips and lumbar region of the spine and a hunched basic position, can certainly be considered as a sport in which it would be justifiable to conduct a research on effects of a long-term regular training on a players' posture.
However, just assessing the Interlimb and trunk asymmetry in the frontal plane of table tennis players won't provide a complete picture of a player's postural deviations. To provide more information about incidence of incorrect postural alignments the sagittal plane should be addressed as well. Table tennis players constantly stand in a hunched basic position while playing. Therefore, it would certainly be more valuable to check does that playing position, maintained for a long time during a player's career, is likely to increase the probability incidence of improper (kyphotic) postural alignments, than to have information about the difference between limb circumferences, which, even in normal population, are always expected to be in favour of a dominant hand.
To summarize the above said. Although the research question of the study is clear it is not quite understandable which meaningful scientific and practical impact the obtained information about Interlimb and trunk asymmetry are expected to provide.

Methods were properly described with sufficient details provided for anyone interested to understand and replicate the research.
However, the sample in the part of table tennis players is really problematic and can't be considered as good representative of the group by any means. (Participants, 130 - 141)
Firstly, it can't adequately represent the population of professional table tennis players, as even without having precise information about the history (quality, regularity and intensity) of the female players training process, from the raw data provided one can tell that girls who are still playing in the cadet (U 14) category, with some of them having just 4-6 years of training behind them, can't be considered as professional TT players. First two years of training are just a play and later in younger cadet days the intensity of training is still not that high. So, the pure duration and intensity of the training process at participants of that young age is still not expected to produce any significant impacts on the posture, which can be related to regular TT training.
Secondly, and more importantly, the already really small sample (N=10) is extremely heterogenous in terms of the growth and developmental age of the female participants representing the TT group. Some of the girls were still in the late puberty or very early adolescent years (13-14 years) at the time of testing, which means within sensitive phase of growth and development, while some of the others are in a biological sense fully grown adult females (20-21 years). As an incidence of growth and development related asymmetries, improper postural alignments or even spinal disorders is proven to be higher in the sensitive growth spurt age, it would be very hard to distinguish the effects of growth and development from the ones caused by a regular TT training on the female players posture. Therefore , I would suggest for the sample to include only female players that are over 16 years of age.

Line 133. ... an average training experience for the TT group was presented by the authors to be of 7 ± 4.3 years. However, according to the data presented in the Raw data Excel table the average training experience should be 9,3 years !!

Lines 175-178 Few words about how to reduce potential measurement/marking error would be beneficial for the study, as marking the points on the subject's body with a black, washable marker is a process that requires anatomical knowledge and experience, as not all anatomical spots are equally observable in different individuals.... and Photogrammetric methods accuracy depends on it.

Validity of the findings

The Table 2. would benefit if the angle and linear asymmetry indices were provided aside to the presented variables' scores with e.g. letters H -high; M-medium, as it would be easier to follow the table results without having to go constantly back to the Methods section to check the values presented for each level of asymmetry.

Lines 252-254 "The control group was characterized by no asymmetry in the KNT parameter and medium (moderate) asymmetry in every other parameter" - this statement is not true as from the presented results it is visible that the values of the "TT- difference in the height of the waist triangles" parameter points to the "severe asymmetry" for the control group.

In the Discussion section the authors' interpretation of the obtained results is not really based on the data results they provided within the Tables 2 and 3.
The authors seem to show bias towards their main research hypothesis "that female table tennis players show greater asymmetry than the comparison group", although the provided results actually do not support it.
e.g. the authors say that "it can be seen that the group of competitive women is characterized by a more significant number of parameters that indicated the occurrence of asymmetry of the trunk in the frontal plane, although a statistical difference was found only in the KSM parameter".
Firstly, only the significant differences really matters and should be used as a scientific evidence of the existing differences between the groups... and from the 3 observed parameters dealing with torso asymmetries (Table 2.), on which significant differences were found between the TT and Controls, on two out of three of those differences the values are higher for the Control group, meaning the asymmetries are more emphasised in the Controls!!
So, the fact that some of the numerical values on 5 out of 9 parameters dealing with torso asymmetries (so just one more parameter on the TT side) show slightly, but not significantly higher values for the TT players than the Controls, does not mean it is possible to point towards the fact that the torso asymmetries are more emphasised in TT players. That claim is not backed up scientifically and therefore is misleading for the readers.

The bias towards their main research hypothesis is also visible in the fact that the authors have put a lot of effort in explaining the only parameter (KSM) on which the higher level of torso asymmetry have been shown by TT players (lines 294-297), but didn't explain at all why on the other two parameters (KNM, UK) the level of asymmetry is higher for the Controls. They should have addressed that finding as well and try to provide an explanation for it.

Basically from the lines 298 - 322 the Discussion section is primarily based on a literature review and theoretical discussion of the topic, but not directly related to the results/findings of the study itself, what should be the main purpose of any Discussion section.

Lines 323-326
The authors say "The differences in the circumferences of the left and right limbs assessed in this study (adopted in this paper as a measure of interlimb asymmetry) were also larger in the examined women who practice table tennis than in those who do not". -
Anyhow, the only statistically significant difference between the groups has been shown for the the right wrist parameter!! ... So, based on the presented results (Table 3.) it can only be stated that both groups had similar sizes compared to the same sides of the body!! Even the differences between the opposite limbs for the TT and Controls are significant on the same variables; dominant and non-dominant hand variables (according to the raw data provided all participants in the study were right handers.) and knees.

As the obtained results were not properly interpreted in the Discussion section of the manuscript that of course led to the fact that claims in the Conclusion section are not fully supported by the results provided.

---

## Round 0.3 · accepted · Accept

Dear authors:

I am pleased to inform you of the acceptance of your manuscript entitled "Interlimb and trunk asymmetry in the frontal plane of table tennis female players" for publication in PeerJ. I appreciate your patience during the review process; sometimes we would like to be faster with it, but, as you know, our main goal is to select the highest quality manuscripts.

I want you to know that you will receive a cordial greeting and congratulations for your contribution to the science of esports.

Dr. Manuel Jimenez

Reviewer 2 ·

Basic reporting

No comment

Experimental design

No comment

Validity of the findings

No comment

Additional comments

The authors have modified the writing with respect to the original submission. The changes allow a better understanding of the document, regarding the problem and the interpretation of results. Some methodological aspects that compromised the quality of the article have also been corrected.

·

Basic reporting

no comment

Experimental design

no comment

Validity of the findings

no comment

Additional comments

no comments